# Advantages and Limitations of Diabetic Bone Healing in Mouse Models: A Narrative Review

**DOI:** 10.3390/biomedicines11123302

**Published:** 2023-12-13

**Authors:** Tanja C. Maisenbacher, Sabrina Ehnert, Tina Histing, Andreas K. Nüssler, Maximilian M. Menger

**Affiliations:** 1Department of Trauma and Reconstructive Surgery, Eberhard Karls University Tübingen, BG Clinic Tübingen, Schnarrenbergstr. 95, D-72076 Tübingen, Germany; THisting@bgu-tuebingen.de (T.H.); MMenger@bgu-tuebingen.de (M.M.M.); 2Siegfried Weller Institute at the BG Trauma Center Tübingen, Department of Trauma and Reconstructive Surgery, University of Tübingen, Schnarrenbergstr. 95, D-72076 Tübingen, Germany; sabrina.ehnert@med.uni-tuebingen.de (S.E.); andreas.nuessler@med.uni-tuebingen.de (A.K.N.)

**Keywords:** fracture healing, bone, mice, diabetes, model, review

## Abstract

Diabetes represents a major risk factor for impaired fracture healing. Type 2 diabetes mellitus is a growing epidemic worldwide, hence an increase in diabetes-related complications in fracture healing can be expected. However, the underlying mechanisms are not yet completely understood. Different mouse models are used in preclinical trauma research for fracture healing under diabetic conditions. The present review elucidates and evaluates the characteristics of state-of-the-art murine diabetic fracture healing models. Three major categories of murine models were identified: Streptozotocin-induced diabetes models, diet-induced diabetes models, and transgenic diabetes models. They all have specific advantages and limitations and affect bone physiology and fracture healing differently. The studies differed widely in their diabetic and fracture healing models and the chosen models were evaluated and discussed, raising concerns in the comparability of the current literature. Researchers should be aware of the presented advantages and limitations when choosing a murine diabetes model. Given the rapid increase in type II diabetics worldwide, our review found that there are a lack of models that sufficiently mimic the development of type II diabetes in adult patients over the years. We suggest that a model with a high-fat diet that accounts for 60% of the daily calorie intake over a period of at least 12 weeks provides the most accurate representation.

## 1. Introduction

Diabetes mellitus is a growing epidemic worldwide [1]. In 2021, 537 million people suffered from diabetes. By 2045, a 46% increase is expected globally, with the highest rates in Africa at 134%. Of the different types of diabetes, type 2 diabetes mellitus (T2DM) is by far the most common, accounting for 90–95% of all cases [2]. T2DM is characterized by peripheral insulin resistance and therefore a relative insulin deficiency. This is in contrast with T1DM, which is caused by an absolute insulin deficiency mainly due to autoimmune destruction of the pancreatic beta-cells. The interaction and combination of genetic, environmental, and metabolic risk factors are known to increase the risk of development of T2DM [3]. Meta-analyses indicate that obesity, unhealthy diet, and low physical activity represent the strongest risk factors for T2DM [4].

Diabetes mellitus is well known for its micro- and macrovascular complications, leading to atherosclerosis, strokes, myocardial infarction, nephropathy, neuropathy, and retinopathy [5]. Another common and well known complication of diabetes mellitus is wound healing disorders caused by different diabetic conditions including neuropathy, impaired angiogenesis, and alterations in inflammatory response [6]. Furthermore, diabetes is associated with an increased risk of fractures [7,8], and represents a major risk factor for the development of non-union fractures [9,10,11]. However, the pathophysiology and underlying mechanisms for diabetic-induced fracture healing failure remain largely unknown [12].

Fracture healing progresses through distinct stages [13], all of which are influenced in diabetic patients [14] (see Figure 1). The initial stage, the inflammation phase, involves the formation of a hematoma infiltrated by various cell types that release cytokines and growth factors. Adequate vascularization during this phase is critical for the subsequent phases of fracture healing [15]. It is noteworthy that diabetes is characterized by a pro-inflammatory state, marked by chronically elevated levels of tumor necrosis factor α (TNF-α), transforming growth factor β (TGF-β), and interleukin-6 (IL-6) [16,17]. Additionally, diabetic patients often experience alterations in angiogenesis and vascularization, which may contribute to impaired fracture healing in this population [18]. During the second stage of fracture healing, endochondral ossification initiates the development of soft cartilaginous tissue, which is subsequently replaced by hard bony tissue in the subsequent phase of fracture healing. Osteoclasts play a crucial role in resorbing the cartilaginous tissue, while osteoblasts gradually replace it with new bone tissue within the callus. Notably, chronic inflammation in diabetic patients resulting from hyperglycemia leads to the production of reactive oxygen species, which is believed to stimulate osteoclast function [19]. There is indeed evidence that diabetes increases the number of osteoclasts, leading to enhanced cartilage resorption [20]. It has been shown that the expression of receptor activator for nuclear factor kB ligand (RANKL) and macrophage colony-stimulating factor (M-CSF) are elevated in diabetic patients [20,21]. TNF-α, RANKL, and M-CSF are known to stimulate osteoclasts and, therefore, the resorption of cartilage and bone. It has been shown that diabetes leads to an increase in osteoclastogenesis [21,22]. Moreover, hyperglycemia leads to an increase in advanced glycation end-products, which inhibit osteoblast function [23]. Additionally, research by Doherty et al. demonstrated that diabetes impairs the regenerative capacity of periosteal progenitor cells [24].

In the final stage of fracture repair, bone remodeling occurs. Once again, osteoblasts and osteoclasts are essential for restoring the stability and trabecular structure of the lamellar bone as it was before the fracture. Therefore, it is highly likely that this remodeling process is also affected in patients with diabetes. 

Regarding the growing number of diabetics, diabetic fracture healing complications following trauma surgery will rise accordingly. The resulting delayed healing and non-union formation is not only associated with significant pain and loss of function for the patient, but also represents a major burden to the health care system [25]. Animal models, especially mouse models, are still the most common and reliable way for preclinical research to analyze and investigate the pathophysiology of fracture healing under diabetic conditions [15,26]. Throughout the literature, different murine fracture and bone healing models are described. A very commonly used model is the closed femur shaft fracture by Bonnarens and Einhorn [27]. However, tibial fractures or even monocortical defect models were used. We examined the described fracture and bone defect models in our relevant literature and discussed the possibilities thoroughly to provide reasonable suggestions for which model to use. All of the relevant literature was screened for the murine diabetes model used. The present review attempts to identify mouse models used to study diabetic fracture healing and evaluates the consistency with human diabetic conditions.

## 2. Search Strategy

For this narrative review, primarily the PubMed database was searched for all types of articles and reviews. The included keywords were “diabetes”, “fracture”, “healing”, and “mice”, leading to 77 results (Figure 2). Abstracts from the earliest available records until June 2023 were included. All publications concerning diabetic fracture healing in mice were included. The abstracts were screened by one operator. Studies were excluded if their manuscripts were not available as full text, as were articles exclusively containing human studies or where no fracture or bone defect model was applied. Forty-five publications turned out to be relevant for the topic, were included and served as the basis of this review. The reference lists of these articles were also reviewed for additional literature and taken into consideration when relevant. The parameters collected were whether the DM model served as a T1DM or T2DM model, the used mouse strain, the age at the start of the treatment leading to diabetes, the time period between the start of the diabetes treatment and the fracture or bone defect, which fracture/bone defect model was used, and the differences found between diabetic and non-diabetic mice in fracture healing. All of the relevant publications were individually searched for the parameters of interest. The results were grouped into three main categories concerning the diabetes model used. 

## 3. Categories of Diabetic Mouse Models

Three major categories of murine models were identified: streptozotocin-induced diabetes models, diet-induced diabetes models, and transgenic diabetes models. Different mouse strains were used (Figure 3, and Table 1 and Table 2).

### 3.1. Streptozotocin-Induced Models

The majority of the studies used a streptozotocin (STZ)-induced a type I diabetes model (Figure 3 and Figure 4, and Table 2). 

Streptozotocin is a broad-spectrum antibiotic causing the destruction of pancreatic beta-cells, which leads to a deficit in endogenous insulin [73]. In our literature research concerning fracture healing, the STZ-induced diabetic model was mostly used in male animals [29,31,36,50] and only a few times in female mice [45,48] (Table 1). Mostly *C57BL/6J* mice were used, whereas few studies used *CD-1* mice or *BALB/c* mice [31,32,40,72]. The injection of STZ was undertaken in a range between 4-week- and 12-week-old mice [29]. The surgery was mostly performed after diabetic conditions were guaranteed for 2–4 weeks [20,32,34,47].

The applied doses of STZ varied. Some authors used a 150 mg/kg body weight single shot intraperitoneal injection (i.p.) [24,36], whereas others used a consecutive i.p. application over 5 days of 40 mg/kg body weight [37,40] or 50 mg/kg body weight, respectively [39,42]. One group injected STZ for 2 days with a concentration of 100 mg/kg body weight [43]. The mice were mostly staged for diabetes 2 weeks after STZ injection. Surgery was usually performed 4 weeks after the start of the injections. Measurement of the blood glucose 1 week after the first STZ injection and 2 weeks after surgery was found in only one study [47]. Different non-fasting blood glucose levels were considered to be diabetic. The levels ranged 200 mg/dL–400 mg/dL [32,36,37,39,45,50]. The lowest blood glucose levels considered diabetic were >220 mg/dL tested on two consecutive days [30] or >220 mg/dL tested 1 week after the last of 5 days of i.p. injection of again 50 mg/kg body weight STZ [49].

Various studies demonstrated that STZ-induced diabetes resulted in a delay in fracture healing [48]. This was associated with a significant reduction in bone density, smaller callus size, lower mechanical strength, and decrease in osteoblast markers [30]. Furthermore, the callus volume was found to be significantly smaller in the STZ-treated mice. This was associated with reduced stiffness, toughness, and maximum torque, describing an impaired mechanical strength compared with wild-type mice [34]. Moreover, STZ-induced T1DM resulted in a reduced amount of cartilage and an increased number of osteoclasts within the callus tissue [20,37,40]. In addition, mesenchymal stem cells were significantly reduced within the fracture callus whereas TNF-α levels were increased [32]. Periosteal cells harvested from STZ-induced T1DM mice showed a decrease in periosteal stem cells and impaired osteogenic potential [24].

Further literature research revealed that male mice were described to be more reliable in developing diabetes and the reliability varied between strains [74,75]. *DBA/2* and *C57BL/6* were described to develop the highest blood glucose levels and, therefore, are considered to be the most suitable strains for STZ-induced diabetic models.

Streptozotocin serving as a type 1 diabetes model was the most used model in research concerning diabetic fracture healing. However, regarding the distribution of type 1 and 2 diabetes worldwide, with more than 90% of the patients suffering from type 2 diabetes, a model reflecting this majority seems to be the more obvious choice. Therefore, it remains questionable if a streptozotocin-induced diabetic fracture model mimics the pathophysiology underlying the impaired fracture in diabetic trauma patients. Accordingly, when applying this model, the translation from preclinical research to clinical practice has to be considered inadequate. On the other hand, a diabetic model simulating the pathogenesis of type 2 diabetes may be more suitable—gradual development of insulin resistance over years, starting with obesity and the onset of a metabolic syndrome, eventually leading to type 2 diabetes, which is associated with additional comorbidities such as cardiovascular diseases and non-alcoholic fatty liver disease [76].

### 3.2. Diet-Induced Models

A less commonly used model to mimic diabetic bone healing in mice represents a diet-induced diabetes model by feeding a high-fat-diet (HFD). Only male *CD57BL/6J* mice have been used [60,63] (Figure 3 and Figure 5, and Table 2). 

Most of the diet-induced diabetes mellitus models used an HFD containing 60% kcal fat [58,62,63] (Table 1). One group of researchers studied a diet containing 45% kcal fat [59,61]. To confirm glucose intolerance, glucose tolerance tests [63] and/or insulin tolerance tests [59] were mostly performed. Different time periods for feeding an HFD diet were seen. In most studies, the mice were 5–8 weeks old when an HFD was initiated. After being fed the HFD, fracture was usually induced after 12–14 weeks. However, a feeding period of 6–8 weeks [61] or using an HFD containing 45% kcal fat was also found [59].

Glucose tolerance and insulin tolerance testing were performed before inducing the 45% kcal HFD, as well as 12 and 13 weeks after the start of the diet [59]. The HFD led to a significant (*p* < 0.001) increase in body weight, a significant (*p* < 0.05) increase in lean body mass, and a significant (*p* < 0.001) increase in fat mass in the mice. Impaired glucose and insulin tolerance were shown, but there was no statistical significance in the differences between the groups. The model was described by the authors as a type-2-diabetes-like phenotype.

After being fed a high-fat diet (HFD) containing 60% of their daily calorie intake from fat for 12 weeks, the mice exhibited a significant increase in body weight [63]. Interestingly, when subjected to a glucose tolerance test, mice fed the 60% calorie HFD showed significantly impaired blood glucose tolerance compared with the control group, in contrast with the 45% calorie HFD-fed mice. While there was a noticeable trend towards reduced vascularity in the callus, the results did not reach statistical significance. However, at 4 weeks post fracture, significant changes were observed. There was a notable reduction in the proportion of woven bone in the callus, along with a decrease in the osteoblast-occupied bone surface. Additionally, a decline in torsional rigidity and an increased presence of fat tissue within the callus were noted. These findings primarily emerged during the later stages of bone healing, specifically at 3 to 4 weeks after the fracture. Notably, after 2 weeks, differences were either absent or not statistically significant. 

Six-week-old mice that were fed a 60% kcal HFD for 6–8 weeks since fracture/surgery showed a significantly lower amount of cartilaginous tissue and less bony tissue within the callus, resulting in delayed fracture repair [61].

The studies relevant to this review exhibited a wide range of variations in terms of the age at which feeding began and in the composition of the diets. Further exploration of the literature revealed conflicting information about the successful induction of diabetes through high-fat diets in specific sexes and mouse strains. 

Pettersson et al. delved into the potential sex-related differences in 11–12-week-old male and female *C57Bl/6* mice by subjecting them to a high-fat diet (HFD) comprising 60% of their calorie intake from fat for 14 weeks [77]. Both sexes displayed a similar increase in body weight compared to the control group. However, the male mice exhibited rising blood glucose levels, while the female mice did not. The serum triglycerides saw a significant increase in both the male and female mice, but the male mice had higher serum cholesterol levels than their female counterparts. After 14 weeks, glucose tolerance was impaired, with this effect being more pronounced in male mice. Hyperinsulinemia and insulin resistance were also only evident in HFD-fed male mice. Furthermore, low-grade, systemic inflammation was stimulated, indicated by an increase in IL-6 and mKc (a murine keratinocyte-derived chemokine similar to human interleukin-8) expression and an increased number of macrophages within the visceral adipose tissue in obese male mice, but not in female mice.

Histing et al. deviated from the norm by feeding *C57BL/6J* mice an HFD containing 60% kcal [78]. Interestingly, in contrast with most publications, Histing et al. found no significant differences in bone density [78]. However, their study reported an increase in adipocytes in bone, although they did not perform a glucose tolerance test. In particular, higher levels of leptin and IL-6 were found, suggesting an increased overall state of inflammation.

Li et al. compared HFD effects on different mice strains, i.e., *Kunming* mice, *C57BL/6, BALB/c*, and *ICR* mice [79]. In every group, half of the animals were fed a 53% kcal HFD and the other half were fed a standard diet for 10 weeks. The *Kunming* mice showed a significantly higher body weight for the HFD-fed mice beginning from the second week of diet, whereas the *ICR* mice demonstrated a higher body weight only from week 6 of the diet. Interestingly, the liver index was significantly higher (*p* < 0.05) in the *C57BL/6*, *BALB/c*, and *ICR* mice. The *Kunming*, *C57BL/6*, and *ICR* mice showed higher peak glucose levels and prolonged periods for lowering the glucose levels in oral glucose-tolerant tests. The insulin tolerance tests showed lowered blood glucose levels in both groups of each mouse strain. However, the area under the curve was significantly higher in all of the HFD-fed groups. In all four mouse strains, no significant differences in inflammatory cytokines (IL-1β, IL-6, IL-10, TNF-α, and MCP-1), LPS, and insulin levels between the control group and HFD fed mice were shown. The group concluded that body weight itself was not reliable as a marker of a diabetic state.

Nishikawa et al. conducted a study in which they fed both 4-week-old and 52-week-old *C57BL/6J* and *BALB/cA* mice a high-fat diet (HFD) consisting of 57.5% of their calorie intake from fat for 9 weeks, including both male and female mice [80]. Glucose tests were performed 5 and 9 weeks after starting the HFD. In the case of young female *BALB/cA* mice, there was no significant increase in body weight. Middle-aged female *C57BL/6J* mice exhibited a slightly higher body weight, but the difference between the standard-fed mice was not statistically significant. Young *C57BL/6J* and *BALB/cA* mice of all ages showed a significant increase in body weight compared with mice fed a standard diet. All HFD groups displayed a notable increase in total fat weight and fat-to-body weight ratio. Both male and female groups of young *C57BL/6J* mice, except for one measurement point, exhibited significantly higher blood glucose levels during the oral glucose tolerance test. Young *BALB/cA* mice showed significantly higher blood glucose levels at one point (male) or two points (female) out of the five measurement time points. In middle-aged *C57BL/6J* mice, the female animals showed higher blood glucose levels three times, while the male animals showed higher levels only once. Similar results were observed for both week 5 and week 9 after starting the diet. The authors concluded that young female *BALB/cA* mice demonstrated resistance to obesity induced by HFD compared with the male animals. Young males of both *C57BL/6J* and *BALB/cA* strains became equally obese. The most severe hepatic lipid accumulation was observed in middle-aged *C57BL/6J* mice of both sexes, followed by young male *BALB/cA* and young male and female *C57BL/6J* mice. Middle-aged *C57BL/6J* mice were more susceptible to HFD-induced obesity compared with the young *BALB/cA* mice. 

Surwit et al. compared different *B6* sub strains: *C57BL/6J* and *A/J* [81,82]. Only the *C57BL/6J* strain developed diabetes with high blood levels of fasting glucose and insulin. Two studies were found concerning the comparison of the *B6* strains: *C57BL/6J* and *C57BL/6NJ*. Nicholson et al. found that both were sensitive to diet-induced obesity (DIO); however, the *B6/J* strain showed a significantly higher body weight accompanied by higher non-fasting serum glucose after 20 weeks. The group stated that, in general, both strains were highly susceptible to DIO [81,82]. Fisher-Wellmann et al. also compared the metabolic alterations in *C57BL/6J* and *C57BL/6NJ* sub strains [83]. They found that both strains were highly susceptible to diet-induced glucose intolerance and metabolic disease; however, *6NJ* mice showed a higher glucose intolerance than the *6J* animals. Regarding insulin resistance, *6J* mice were resistant to glucose-stimulated insulin secretion and were partially protected from diet-induced fasting hyperinsulinemia. They concluded that the *C57BL/6N* sub strain might be the better model for their study purposes.

In summary, the majority of these studies consistently identified male *C57BL/6J* and *C57BL/6N* mice as being the most susceptible to the development of a diabetic state, characterized by increased inflammation, hyperinsulinemia, and insulin resistance. However, it is worth noting that not every study fully supported this assumption, as some failed to demonstrate rising blood glucose levels [78] or an increase in inflammatory cytokines [79]. It is important to highlight that the existing literature suggests that a high-fat diet comprising 60% of daily calorie intake from fat for a minimum of 12 weeks is required for the successful induction of diabetes mellitus [63].

### 3.3. Transgenic Models

The most common transgenic mouse models for obesity and diabetes research are *ob/ob* (*Lep^ob^*) and *db/db* (*Lepr^db^*) mice, with the *C57BL/6J* strain (Figure 3 and Figure 6, Table 2). 

*Ob/ob* mice are characterized by a mutation of the gene encoding for leptin, leading to a lack of circulating leptin [84]. *Db/db* mice, on the other hand, are characterized by a deficiency in the leptin receptor. Both mutations affect the feeling of satiety. As a result, *db/db* and *ob/ob* mice develop severe obesity, hyperphagia, and hypometabolism [84,85]. In addition, *db/db* mice suffer from polydipsia and polyuria [84,85].

Leptin-deficient *ob/ob* mice were used in only one study [55], whereas leptin receptor-deficient *db/db* mice were the main subjects in recent research [64,65,66,86] (see Table 1). For instance, Wagner et al. compared the effects of transgenic diabetic mice with sympathectomized and wild-type animals [65]. In particular, they discovered striking similarities in diabetic and sympathectomized mice, leading to the hypothesis that diabetic neuropathy directly impedes the process of fracture healing. This hypothesis was further supported by their finding that treatment with a beta3-adrenergic agonist improved bone healing in both groups, counteracting the negative effects of diabetes. 

Wallner et al. revealed various impairments in transgenic diabetic mice, including reduced osteoblast proliferation, migration, differentiation, and angiogenesis [86]. Furthermore, *db/db* mice exhibited delayed periosteal mesenchymal osteogenesis, premature apoptosis of the cartilage callus, impaired microvascular invasion, and increased serum concentrations of TNF-α [64,67]. Noteworthy, neither of the studies confirmed a diabetic state by testing glucose nor an insulin tolerance or blood glucose levels.

However, leptin, which is produced by adipocytes, is known to have an influence on bone metabolism [87]. Human stromal cells were shown to express leptin receptors and presence of leptin favored differentiation of mesenchymal stem cells into osteoblasts while inhibiting their differentiation into adipocytes [88]. Leptin was shown to increase bone formation [89] and it was shown that obesity led to higher bone mineral density in the spine, but lower density in the shorter femora [90]. In addition, leptin influenced osteoclasts and bone resorption [91], which, overall, led to alterations in bone metabolism and remodeling. 

Therefore, the changes in bone healing might not only be caused by the development of diabetes, but also by the changes in the leptin knockout. In fact, it has to be considered that leptin knockout influences the bone development of mice during adolescence.

Tevlin et al. analyzed the bones and fracture healing in 4-week-old *ob/ob* mice before the onset of diabetes mellitus [64]. Mechanical strength after fracture healing and the number of skeletal stem cells (SSCs) and bone, cartilage, and stromal progenitors (BCSPs) were investigated. Wild-type and *ob/ob* mice did not show significant differences to the wild-type controls before the onset of diabetes. Accordingly, the authors concluded that impaired fracture healing was not related to aberrant leptin signaling. Even later, when the diabetes manifested, there were no significant differences in bone mechanical strength or SSC and BCSP cell numbers observed between *ob/ob* and wild-type mice. In contrast, the mechanical strength was significantly impaired in the DIO (*p* < 0.05) and STZ-induced (*p* < 0.0001) diabetes model. Both models showed significantly reduced numbers of SSCs (DIO: *p* < 0.05; STZ: *p* < 0.05) and BCSPs (DIO: *p* < 0.05; STZ: *p* < 0.01) in the callus 1 week after fracture.

Despite these findings, it remains questionable whether deficiency in the leptin or leptin receptor directly influences bone healing, especially concerning the effects on bone remodeling, which takes place after final development of the bones and in the last step of fracture healing. In addition, reduced height of the tibial growth plate and fewer chondrocyte columns in *ob/ob* mice compared with the control group was demonstrated [92]. This underlines and strengthens the hypothesis of possible changes in bone development in *ob/ob* and *db/db* mice before reaching the adult bone stage, which might likely influence bone healing afterwards. Therefore, further studies are required to confirm *ob/ob* mice as a reliable mouse model to investigate bone healing in diabetes. 

Further transgenic mouse strains include the *MKR* mice (*FVB/N* background) or *Akita* mice (*C57BL/6J* background). *MKR* mice developed a non-obese phenotype with reduced insulin production [93]. These mice showed delayed fracture healing and a higher pro-inflammatory state compared with the wild tape controls [29]. *Akita* mice were characterized by a mutation in one allele of the insulin-2 gene [69]. The mice showed significantly less callus with reduced cartilage and bone formation.

**Table 1 biomedicines-11-03302-t001:** Overview of murine fracture diabetes models.

	Model	Diabetes Type	Mouse Strain	Diabetes Symptoms	Considered Diabetic	Treatment	Surgery	Fracture Healing	Ref.
Medical-induced	STZ *	T1DM	*C57BL/6J* *CD-1* *BALB/c*	high non-fasting blood glucose level	>200 mg/dL ≥ 400 mg/dL blood glucose level	low dose: 40–50 mg/kg body weight 5 days (i.p.) high dose: single shot 150 mg/kg body weight (i.p.)	2–4 weeks after injection	delayed fracture healing	[20,24,28,29,54,64]
Diet-induced	HFD *	T2DM	*C57BL/6J*	obesity impaired glucose tolerance hyperglycemia	significant larger AUC * in GTT */ITT *	45–60% kcal fat	(6-) 8–14 weeks after start of HFD	delayed fracture healing, reduced callus size, reduced torsional rigidity, increased callus adiposity	[28,58,59,60,61,62,63,64]
transgenic	*ob/ob*	T1DM	*C57BL/6J* *Lep^ob/ob^*	severe obesity hyperphagia hypometabolism	no testing	-			[55]
	*db/db*	T1DM	*C57BL/6J* *Lep^db/db^*	severe obesity hyperphagia hypometabolism polydipsia polyuria	no testing	-	12–14 weeks old	impaired osteoblast invasion, proliferation, and differentiation; impaired angiogenesis; decreased osteoblast invasion	[65,66,67,68,71]
	Akita	T1DM	*C57BL/6J* *(C57BL/6-INS2^Akita^/J)*	hypoinsulinemia	no testing	-	18 weeks old	reduced callus, less cartilage and bone	[69]
	MKR	T1DM	*FVB/N*	hyperglycemia	Blood glucose > 250 mg/dL	-	12 weeks	-	[56]

* STZ: streptozotocin; HFD: high-fat diet; AUC: area under the curve; GTT: glucose tolerance testing; ITT: insulin tolerance testing.

**Table 2 biomedicines-11-03302-t002:** Summary of the relevant literature.

DM *	Model	Model Specifi-Cation	Mouse Strain	Sex	Age at Start of Diabetic Treatment	Time Until Intervention/Length of Diet	Testing	Fracture Model	Alterations in Diabetic Fracture Healing	Ref.
T1DM *	STZ *	40 mg/kg on 5 consecutive days	*CD-1*	male	9 weeks	3 weeks	blood glucose testing: 300–550 mg/dL (mean 411 mg/dL), considered diabetic: two consecutive measurements > 250 mg/dL	closed transverse tibia fractures, intra-medullary pin	12 days after fracture: callus size similar, 16 days: significantly smaller callus, 78% more osteoclasts	[20]
T1DM	STZ	150 mg/kg single shot	*C57BL/6J* *(Cg-Tg(Col1a1*2.3-GFP)1Rowe/J)*	both	4–8 weeks	4 weeks	weight and fasting blood glucose: significant higher fasting blood glucose, no significant weight difference	closed femur fracture, pin fixation	periosteal cell deficient in osteogenic differentiation, reduced population of periosteal mesenchymal progenitors, reduced proliferation capacity	[24]
T1DM	STZ	50 mg/kg on 5 consecutive days	*C57BL/6J*	male	12 weeks	16 days	daily weight and blood glucose levels monitoring for 16 days	open transverse femur fractures, pin stabilization, 0.5 mm fracture gap, clip stabilization	significantly elevated TNF-α, IL-1β, COX2, and NOS-2 levels, delayed bone defect healing	[29]
T1DM	STZ	40 mg/kg on 5 consecutive days	*C57BL/6J*	both	9 weeks	3 weeks hyperglycemic	considered diabetic when blood glucose levels exceed 220 mg/dL for 2 consecutive measurements	closed transverse femoral shaft fractures, pin stabilization	inhibition of ciliary gene expression, delayed fracture healing, significantly reduced bone density and mechanical strength, reduced osteoblast marker expression and decreased angiogenesis	[30]
T1DM	STZ	40 mg/kg on 5 consecutive days	*CD-1*	male	8 weeks	3 weeks hyperglycemic	mean glucose values of 25–28 mmol/L (vs. 6–8 mmol/L)	transverse closed femur fractures, pin stabilization	reduced VEGFA expression, reduced angiogenesis in areas of endochondral ossification	[31]
T1DM	STZ	40 mg/kg on 5 consecutive days	*CD-1*	male	8 weeks	3 weeks diabetic	blood glucose levels > 250 mg/dL considered diabetic	transverse closed femur fractures, pin stabilization	increased TNF-α levels and reduced mesenchymal stem cell numbers in new bone areas	[32]
T1DM	STZ	40 mg/kg on 5 consecutive days	*BALB/c*	male	12 weeks	1 week, only the diabetic mice where used	blood glucose levels > 270 mg/dL, mean blood glucose level was 300 mg/dL	mid-diaphyseal femur fracture model	no non-diabetic control group	[33]
T1DM	STZ	50 mg/kg on 5 consecutive days	*C57BL/6J*	n.n.	8 weeks	3 weeks diabetic	considered diabetic with blood glucose levels > 220 mg/dL	closed femur fracture, pin fixation	reduced mechanical strength of fracture callus 35 days after fracture	[34]
T1DM	STZ	150 mg/kg single shot	*C57BL/6*	male	6–8 weeks	n.n.	blood glucose levels 250 mg/dL	open tibia fracture, pin stabilization, 2mm diameter defect of the calvarium		[35]
T1DM	STZ	150 mg/kg single shot	*C57BL/6*	male	12 weeks	2 weeks	blood glucose levels > 400 mg/dL	closed transverse femur fracture, pin stabilization	2 and 3 weeks after fracture: smaller callus, significantly reduced osteoclast size but elevated numbers, no alterations ins osteoblast function	[36]
T1DM	STZ	40 mg/kg on 5 consecutive days	*C57BL/6*	male	8 weeks	3 weeks diabetic	considered diabetic with blood glucose levels > 250 mg/dL	transverse tibial and femoral shaft fracture, pin stabilization	three times higher blood glucose levels, decrease in callus and cartilage area, higher TNF-α levels, increase in chondrocyte apoptosis	[37]
T1DM	STZ	40 mg/kg on 5 consecutive days	*C57BL/6 (Col2α1Cre^−^.FOXO1^L/L^)*	n.n.	12–14 weeks	3 weeks diabetic	considered diabetic with blood glucose levels > 220 mg/dL for two consecutive tests	closed femoral shaft fracture, pin stabilization	three times increase in osteoclasts, two−three times increase in RANKL mRNA and RANKL expressing chondrocytes	[38]
T1DM	STZ	50 mg/kg on 5 consecutive days	*C57BL/6J*	male	7 weeks	3 weeks	considered diabetic with non-fasting blood glucose levels > 300 mg/dL, diabetic: mean 493 mg/dL vs. control: mean 140 mg/dL	drill hole injury, round defect of 1 mm diameter, no stabilization	delayed bone healing at day 7 and 10	[39]
T1DM	STZ	40 mg/kg on 5 consecutive days	*CD-1*	male	8 weeks	3 weeks diabetic	considered diabetic with blood glucose levels > 250 mg/dL	closed transverse femoral fracture, pin stabilization	significantly reduced callus size at day 16 and 22, on day 10 just missed significance (*p* = 0.07)	[40]
T1DM	STZ	40 mg/kg on 5 consecutive days	*C57BL/6J*	male	6–8 weeks	n.n.	considered diabetic with fasting blood glucose levels > 11.1 mmol/L	closed femoral fracture,	significantly reduced bone mineral density, trabecular number/separation/thickness (unfractured bone)	[41]
T1DM	STZ	40 mg/kg on 5 consecutive days	*CD-1*	male	8 weeks	3 weeks hyperglycemic	considered diabetic with blood glucose levels > 250 mg/dL	closed transverse femoral shaft fracture, pin stabilization	upregulation of several chemokines, chondrocytes showed enhanced CCL4 expression	[42]
T1DM	STZ	100 mg/kg on 2 consecutive days	*C57BL/6*	male	10 weeks	4 weeks	considered diabetic with glucose levels > 290 mg/dL doses, testing 2 weeks after STZ injections	monocortical tibial defect, 0.8 mm in the anterior cortex	low VEGF and Bmp2/4 expression in bone and impaired bone regeneration	[43]
T1DM	STZ	50 mg/kg on 5 consecutive days	*C57BL/6*	male	n.n.	1 week	decrease in body weight, significant increase in glucose levels	femoral mono-cortical bone defect: 4 mm length, 1 mm diameter	delay in bone regeneration, large areas of loose connective tissue within the defects, reduced expression of osteonectin	[44]
T1DM	STZ	50 mg/kg on 4 consecutive days	*C57BL/6*	female	10 weeks	4 weeks	considered diabetic with non-fasting blood glucose levels > 300 mg/dL, measurement 4 days after last injection, decreased body weight	femoral bone defect, 0.9 mm drill	delayed bone repair (controls healed within 7 days, diabetics not), significantly lower ratio of RANKL/OPG	[45]
T1DM	STZ	40 mg/kg on 5 consecutive days	*CD-1*	male	8 weeks	3 weeks diabetic	considered diabetic with blood glucose levels > 250 mg/dL	closed transverse tibia and femur fracture, pin stabilization	upregulation of 31 out of 38 tested inflammatory gene sets at day 16, significantly increased the number of TNF-α positive proliferative and hypertrophic chondrocytes on day 16	[46]
T1DM	STZ	50 mg/kg on 5 consecutive days	*C57BL6/J*	male	6 weeks	2 weeks	considered diabetic with blood glucose levels > 300 mg/dL	closed femoral shaft fracture	reduced bone volume to total bone volume ratio and trabecular thickness in lumbar vertebrae, decreased callus mineralization at 6 weeks	[47]
T1DM	STZ	50 mg/kg on 4 consecutive days	*C57BL6*	female	10 weeks	4 weeks	considered diabetic with non-fasting blood glucose levels > 300 mg/dL, testing 4 days after last injection	femoral bone defect, drill defect, 9 mm diameter of the drill	significantly delayed defect healing at days 7 and 10	[48]
T1DM	STZ	50 mg/kg on 5 consecutive days	*C57BL/6J*	both	8–10 weeks	4–6 weeks diabetic	considered diabetic with blood glucose levels > 220 mg/dL	closed transverse femur shaft fracture, pin stabilization	enhanced RANK activation in periosteal cells, loss of skeletal stem cells	[49]
T1DM	STZ	50 mg/kg on 5 consecutive days	*C57BL/6J*	male	8 weeks	4 weeks	considered diabetic with blood glucose levels >300 mg/dL, testing 1 week after last injection	closed femoral fracture, nail stabilization	6 weeks: significant reduction in bone formation, less bone mass, low bone density, porous woven bone, 54% decreased bone volume fraction, 39% decreased bone connectivity density	[50]
T1DM	STZ	50 mg/kg on 5 consecutive days	*C57BL/6J*	male	9 weeks	3 weeks diabetic	considered diabetic with blood glucose levels > 13 mM for 3 consecutive weeks	closed femur shaft fractures, pin stabilization	significantly elevated inflammation-related biomarkers at days 2 and 7, more markers were elevated at day 2	[51]
T1DM	STZ	50 mg/kg on 5 consecutive days	*C57/B6*	male	8–10 weeks	3 weeks diabetic	considered diabetic with blood glucose levels > 13 mmol/L for 3 consecutive weeks	closed transverse femur shaft fractures, pin stabilization	no non-diabetic control group	[52]
T1DM	STZ	50 mg/kg on 4 consecutive days	*C57BL/6J*	female	8 weeks	3 weeks	considered diabetic with non-fasting blood glucose levels > 300 mg/dL, decrease in body weight	femoral bone defect, 0.9 mm diameter drill	reduced number of macrophages on day 2 but not day 4, slight increase in TNF-α mRNA levels at the defect site	[53]
T1DM	STZ	40 mg/kg on 5 consecutive days	*CD-1*	n.n.	n.n.	2 weeks diabetic	considered diabetic with blood glucose level > 250 mg/dL	marrow ablation in the proximal tibia		[54]
T1DM/ T2DM *	STZ/ HFD *	40 mg/kg on 5 consecutive days/ 45% kcal fat	*C57BL/6J*	male	8 weeks	12 weeks	glucose tolerance test, impaired glucose tolerance and hyperinsulinemia in HFD mice	tibial cortical bone defect, 0.8 mm diameter	delayed bone healing, increase in reactive oxygen species, inhibitory effects on osteoblasts	[28]
T2DM	HFD	60% kcal fat	*C57BL/6J*	male	5 weeks	7 weeks	significantly elevated fasting blood glucose levels, pre-diabetic hyperglycemia and glucose intolerance	closed tibial shaft factures, pin stabilization	significantly lower bone strength at day 35, significantly lower bone volume, bone volume density and bone mineral density at days 21 and 35, pathological accumulation of AGEs * in callus leading to increased collagen-fiber crosslink density	[58]
T2DM	HFD	45% kcal fat	*C57BL/6J*	male	6 weeks	14 weeks	glucose tolerance test and insulin tolerance test, significant higher body weight, impaired glucose and insulin tolerance in mice	2 mm femoral diaphyseal defect, pin stabilization	no significant differences	[59]
T2DM	HFD	45% kcal fat	*C57BL/6J*	male	8 weeks	12 weeks	glucose tolerance test and insulin tolerance test, higher net blood glucose level and total insulin release during the test, significantly larger area under the curve	2mm intercalary segment from the femoral diaphysis, pin fixation	impaired bone healing	[60]
T2DM	HFD	60% kcal fat	*C57BL/6J*		6 weeks	6–8 weeks	no testing	open transverse osteotomy in the mid-diaphysis of the femur, pin stabilization	significantly lower percentage of cartilaginous callus area in total callus area at 1 week, less bony callus at 2 weeks, callus and fracture lines still visible at 4 weeks	[61]
T2DM	HFD		*C57BL/6J*	male	6 weeks	-	2.4-fold increase in fasting blood glucose	tibia fracture, pin stabilization	Significantly more cartilaginous and adipose tissue in fracture callus, altered osteogenesis and chondrogenesis, decreased blood serum osteocalcin	[62]
T2DM	HFD	60% kcal fat	*C57BL/6J*	male	5 weeks	12 weeks	increased weight and impaired glucose tolerance	open tibia fractures, pin stabilization	significantly increased fracture callus adiposity at days 21, 28, and 35; significantly decreased woven bone at day 21; significantly reduced torsional rigidity at y 35	[63]
T1DM/ T2DM	TG * (STZ, HFD)	*db/db*	*C57BL/6*	female	10 weeks and 4 weeks	-	no testing	open femoral shaft fracture, pin stabilization	all models showed significantly reduced strength, db/db: 10-week-old mice: significantly decreased bone mineral density, significantly decreased bone volume/tissue volume; 4-week-old mice: no significant difference in strength	[64]
T1DM	TG	*ob/ob*	*C57BL/6*	n.n.	-	8 weeks	no testing	mid-skull transcortical defects, 3.5 mm drill	diabetic macrophages impair bone regeneration, alterations in vascularization and increased number of adipocytes	[55]
T1DM	TG	*MKR*	*FVB/N*	male	-	8 weeks	blood glucose: diabetic: 350 mg/dL vs. non-diabetic: mean 150 mg/dL	closed femoral shaft fracture, pin stabilization and femoral drill hole model, 0.8 mm diameter drill	less callus formation at days 10 and 16	[56]
T1DM	TG	*db/db*	*C57BL/6J*	female	-	12–14 weeks	no testing	1 mm monocortical tibial defect	reduced bone regeneration, decreased osteoclasts, reduced osteoblastogenesis	[65]
T1DM	TG	*db/db*	*C57BL/6J*	n.n.	-	12–16 weeks	no testing	1 mm monocortical tibial defect	increased activation of TGF-β pathway in callus, significant differences in expression of multiple genes, significantly higher expression of inflammation-associated factors	[66]
T1DM	TG	*db/db*	*C57BL/6*	female	-	12–14 weeks	significant differences in glucose tolerance testing	closed femoral shaft fracture, pin stabilization	no differences at day 3, delayed healing from day 7 on, callus still visible at day 30 vs. healed fracture in controls, poor chondrogenesis, enhanced chondrocyte apoptosis at day 7 and 14	[67]
T1DM	TG	*db/db*	*C57BL/6J*	n.n.	-	12–16 weeks	no testing	1 mm monocortical tibial defect	impaired osteogenesis	[68]
T1DM	TG	*Akita*	*C57BL/6J*	male	-	18 weeks	hyperglycemic at 6 weeks: mean: 496 mg/dL, at 18 weeks: mean: 574 mg/dL	femur fractures	significantly smaller callus with less cartilage and bone area at days 14 and 21, reduced torsional strength	[69]
T1DM	TG	*Agouti*	*Agouti*	n.n.	-	8 weeks	described as hyperglycemia, hyperinsulinemia, glucose intolerance, and insulin resistance by 8 weeks of age	mid-shaft tibia osteotomy, two-ring external fixation spanning	no differences in new bone formation	[70]
T1DM	TG	*db/db*	*C57BL/6J*	n.n.	-	12 weeks	n.n.	femur shaft fractures	decreased bony callus areas, reduced osteoblast numbers, reduced RANKL	[71]

* DM: diabetes mellitus; T1DM: type 1 diabetes mellitus; T2DM: type 2 diabetes mellitus; STZ: streptozotocin; HFD: high-fat diet; AGEs: advanced glycation end products; TG: transgenic.

## 4. Comparison of the Bone Fracture/Defect Models Used

A wide range of different models to induce bone healing was used, including closed fractures, open fractures, and monocortical defects. Closed-shaft fractures were the most commonly used model, induced mainly on the femur [24,36,64] and a few on the tibia [37]. Femoral intramedullary pin fixation was the most commonly used osteosynthetic method [56,64]. Monocortical, mostly tibial, defects between 1 mm and 0.8 mm were used without osteosynthetic fixation [20,65,86].

Some researchers addressed their choice of the model, but the majority did not. The most established model was the fracture model developed by Bonnarens and Einhorn [27], a closed-shaft fracture mostly applied to the femur. One group argued their choice to use a tibial fracture osteotomy model by the potential increase in skeletal fragility of their HFD-fed mice [63]. However, this mimics a bone defect model rather than a fracture healing model.

A wide range of murine fracture models and fracture stabilization methods are described throughout the literature. As mostly found in the analyzed studies, femoral shaft fractures are described to be the most standardized fracture model in mice [26]. Frade et al. introduced a fracture fixation method by using an intramedullary wire [94]. Holstein et al. developed a stable closed femoral shaft fracture by using a screw fixation [95]. Histing et al. showed that this stabilization method led to endochondral fracture healing and callus formation [96]. In an ex vivo study, the mechanical characteristics of different osteosynthesis methods for femoral shaft fractures were analyzed [97]. An external fixation or locking plate fixation of the fracture showed the highest torsional stiffness, followed by the mouse nail. The lowest torsional stiffness was seen in the pin fixation. However, the methods with the highest rotational stability required an open approach to the femur. Regarding closed femoral fracture stabilization, screw fixation showed the highest torsional stiffness. Especially when addressing biological fracture healing and it´s complications, a stable, closed, highly reproducible and standardized fracture model should be chosen.

## 5. Discussion

Our review concerning fracture healing in different diabetic mouse models revealed 77 relevant publications and showed three main groups of mouse models that can be differentiated. Diabetes induced by streptozotocin, transgenic models, mainly *db/db* mice, and high-fat diet-induced obesity/diabetic state. Wide variations within these models, like different STZ doses, different mouse strains, and different diets, were revealed. In each of the models, impaired fracture healing was observed. However, which model is the best to use to simulate the human diabetic state for fracture healing remains unclear and has hardly been discussed. None of the publications discussed their choice of murine diabetic model. Only a few of the authors explained why they chose their fracture or bone defect model. 

However, we were able to show that throughout the literature, diabetic fractures in healing in mice showed delayed and impaired fracture healing in the different mice models. Wei et al. showed significantly elevated TNF-α, IL-1β, COX2 and NOS-2 levels and delayed bone defect healing in diabetic mice [29]. This was supported by Ko et al. who found increased TNF-α levels and reduced mesenchymal stem cell numbers in new bone areas [32]. Kayal et al. showed that within a closed tibia fracture with intramedullary pin fixation, the callus 16 days after fracture was significantly smaller in diabetic mice and contained 78% more osteoclasts in comparison with non-diabetic mice [20]. Doherty et al. showed a reduced population of periosteal mesenchymal progenitors and reduced proliferation capacity [24]. Kashara et al. found that 2 and 3 weeks after fracture, the callus size and the osteoclast size in diabetic mice were significantly reduced, but no alterations in osteoblast function were found [36]. In a different study, Kayal et al. found significantly reduced callus size at days 16 and 22; testing on day 10 just missed statistical significance [37]. Lu et al. showed that even after 35 days, the mechanical strength of the fracture callus was reduced in diabetic mice [34]. In summary, the studies were able to show a prolonged healing time and reduced callus size. The significant alterations were shown from days 10 or 16, indicating that the bone turnover within the later fracture healing stages were mostly affected in callus size and mechanical strength. Within the earlier fracture healing stages, the main differences were elevated inflammatory cytokines [42]. Shimoide et al. found that on day 2 after a bone defect, the macrophages numbers were lower in diabetic mice, but not on day 4 [53]. They concluded that diabetes reduced macrophage accumulation during the inflammatory state of fracture healing. These findings, that a delay in fracture healing occurs within the later healing stages, are supported by the findings that the osteoclasts and their regulation are upregulated. Ko et al. showed an enhanced RANK activation in periosteal cells, as well as a loss in skeletal stem cells [49]. Tamura et al. showed a decreased ratio of RANKL/OPG [45]. This was strongly supported by the findings of Kayal et al. and Rőszer et al. Kayal et al. found that in diabetic mice, 78% more osteoclasts were found, underlining that the bone turnover and remodeling processes were affected by diabetes [20]. Rőszer et al. came to the same conclusion, with no differences found in callus development at day 3 after fracture [67]. However, on day 30, the controls showed complete healing of the fractures, whereas in diabetic mice, the callus was still visible. 

### Limitations

We feel that the presented review provides a good overview of the existing literature and reveals the relevant issues that should be considered when choosing a diabetes and/or a fracture healing model. However, a few limitations have to be pointed out. 

Only pre-clinical studies where included, so no evidence can be provided that the alterations found in diabetic mice really mimic the human diabetic conditions. However, the described changes in murine diabetic fracture healing, like elevation of inflammatory cytokines and delayed fracture healing, have been described in human diabetic patients within the literature [3,6]. Another limit to be mentioned is that only publications concerning fracture healing were included. Another interesting point would be to elucidate how diabetic stage affects bone turnover and remodeling in unfractured bone, and how long a diabetic state should occur until the bone is similar to the bone of diabetic patients.

## 6. Which Model to Choose?

In humans, 90% of patients suffered from T2DM. In contrast with T1DM, T2DM develops with time, mainly in obese patients. Consequently, T2DM manifests mostly in adult or elderly patients, although the international diabetes foundation recently revealed an alarming increase in T2DM in young patients. Therefore, a model respecting these aspects of the human patient cohort seems to be the most logical. Regarding this, the STZ-induced diabetes model representing T1DM does not mimic the majority of patients. T2DM is better presented by the DIO model and *db/db* or *ob/ob* mice. However, especially when considering studies investigating the physiology of bone development and regeneration, transgenic models with *db/db* and *ob/ob* mice may suffer from the influence of the leptin gene on bone physiology.

In our opinion, the chosen diabetes model used should represent the pathophysiology of T2DM. T2DM and metabolic syndrome are mainly characterized by a chronic state of inflammation and onset in adult age. Feeding an HFD highly mimics the pathogenesis of T2DM. In our opinion, there might be possible alternative diets concerning the composition of the food worldwide with high sugar, high fat, and a too high caloric intake overall. Blood glucose testing and/or insulin testing should be performed because the development of obesity itself is not a reliable marker [79].

Regarding the best fracture model, a closed femoral shaft fracture with a screw fixation seems to be the most reproducible, representing endochondral bone healing focusing on the biological fracture healing mechanisms by minimizing the mechanical influence through stable fixation. 

## 7. Conclusions and Recommendation

Obesity and its association with type 2 diabetes (T2DM) represent a growing global health problem. The increasing incidence of fractures and the significantly higher rates of nonunion and delayed fracture healing in diabetic patients highlight the urgent need for further research in this area. In this review, existing mouse diabetes models are critically evaluated to determine the most appropriate model for studying fracture healing in the context of T2DM.

The aim is to select a mouse model that accurately reflects clinical reality, particularly with regard to elderly patients who gradually develop T2DM over time. We suggest that a model with a high-fat diet that accounts for 60% of the daily calorie intake over a period of at least 12 weeks provides the most accurate representation. This choice is underpinned by the following key considerations:High-fat diet: The use of a high-fat diet is known to cause obesity and insulin resistance in mice, similar to the features of T2DM in humans. Prolonged consumption of a high-fat diet helps reproduce the gradual onset of T2DM that typically occurs in older patients.Realistic scenario: we emphasize the importance of modeling the gradual evolution of T2DM over time in accordance with the clinical course observed in patients.Obesity and type 2 diabetes: High-fat diets are a leading cause of obesity, and a known risk factor for type 2 diabetes. Consequently, the use of this nutritional model allows researchers to study the complex interplay between obesity, diabetes, and fracture healing.

It is crucial to emphasize that the selection of a mouse model should be driven by the particular research objectives and the specific aspects of diabetes-related fracture healing under investigation. 

However, a few points should be mentioned to enhance the quality of a publication concerning diabetic fracture healing in mice: The chosen mouse model (fracture and diabetic model) should be described meticulously.Researchers should explain precisely why they chose the specific models.Diabetic testing should be performed with, for example, fasting blood glucose levels, glucose tolerance tests, or insulin tolerance tests.

Different mouse models may be better suited for different facets of diabetes research. Furthermore, ethical and practical considerations related to animal models in research should be carefully weighed to ensure that the chosen model aligns with the research goals and accurately represents the clinical context it aims to replicate. 

## Figures and Tables

**Figure 1 biomedicines-11-03302-f001:**
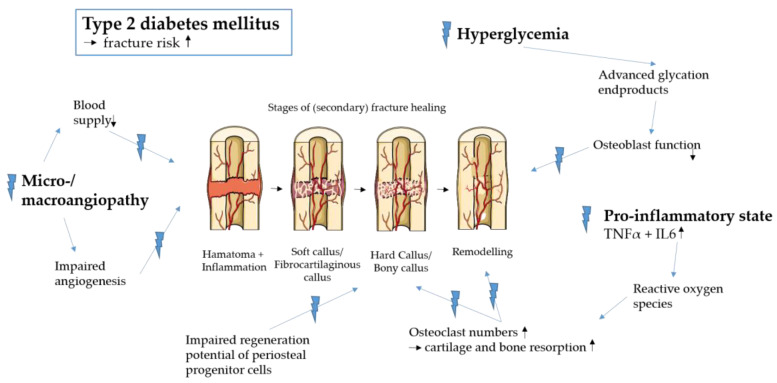
Fracture healing in diabetic patients (provided by Servier Medical Art).

**Figure 2 biomedicines-11-03302-f002:**
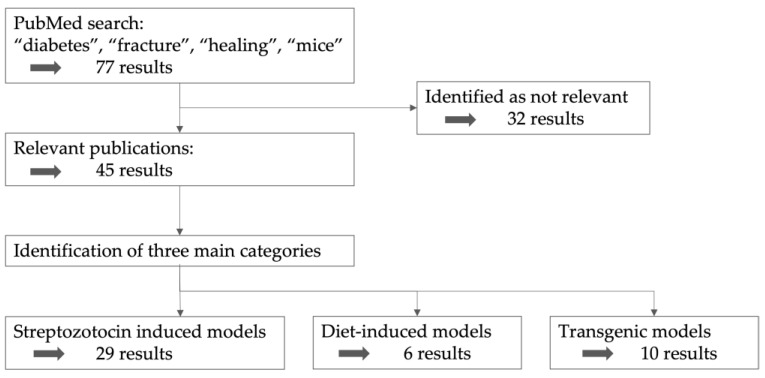
Search strategy.

**Figure 3 biomedicines-11-03302-f003:**
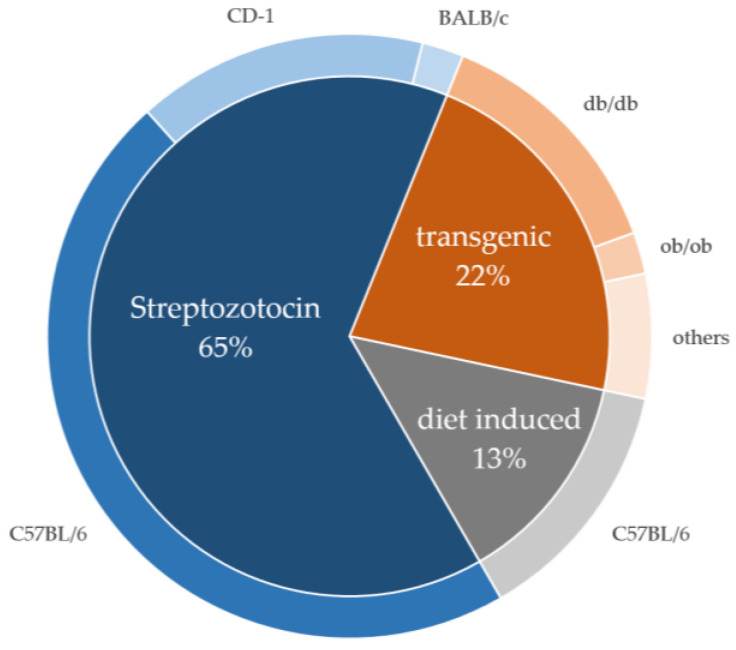
Distribution of the diabetes models among the relevant publications. Three categories of murine diabetes models were identified: streptozotocin-induced, transgenic, and diet-induced models. The distribution is shown in the figure. The mice strains used for STZ-induced diabetes were *C57BL/6* (72%), *CD-1* (24%), and *BALB/c* (3%). The most common used transgenic mice strain was *db/db* mice (60%, *C57BL/6* background). Furthermore, *db/db* (10%) and a few other strains (30%) were used. All of the diet-induced diabetes models used *C57BL/6* mice (100%) [20,24,28,29,30,31,32,33,34,35,36,37,38,39,40,41,42,43,44,45,46,47,48,49,50,51,52,53,54,55,56,57,58,59,60,61,62,63,64,65,66,67,68,69,70,71,72].

**Figure 4 biomedicines-11-03302-f004:**
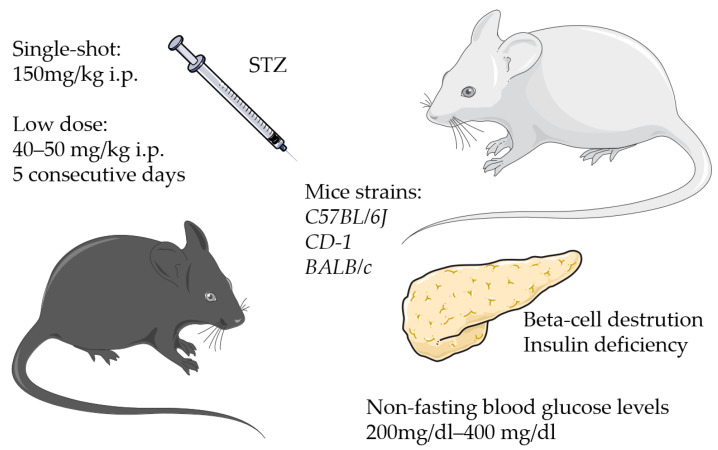
Streptozotocin-induced diabetes (provided by Servier Medical Art).

**Figure 5 biomedicines-11-03302-f005:**
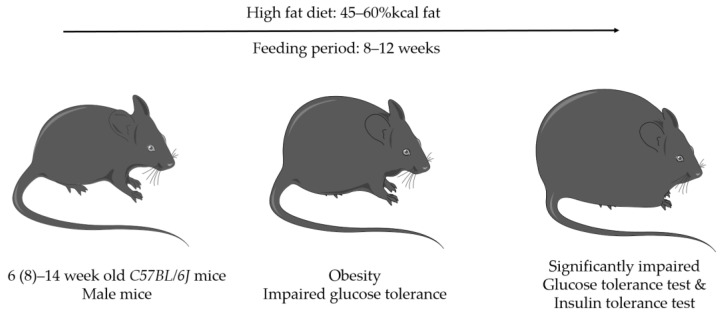
Diet-induced diabetes (provided by Servier Medical Art).

**Figure 6 biomedicines-11-03302-f006:**
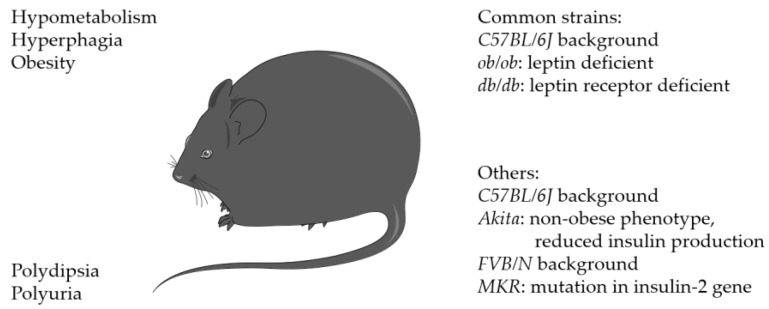
Transgenic diabetes model (provided by Servier Medical Arts).

## Data Availability

Not applicable.

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
