# Peer review of "Advantages and Limitations of Diabetic Bone Healing in Mouse Models: A Narrative Review"

_biomedicines, 2023, doi:10.3390/biomedicines11123302_

Round 1

Reviewer 1 Report

Comments and Suggestions for Authors

This article reviews diabetic bone healing models in mice, focusing on their advantages and limitations. The authors aim to evaluate existing mouse models for the study of diabetic fracture healing and determine the most appropriate model. They categorize models into three types: Streptozotocin-induced, diet-induced and transgenic models, and discuss their characteristics and implications for understanding diabetic fracture healing. The paper suggests that a high-fat diet model most accurately represents type 2 diabetes. The authors emphasize the importance of realistic scenarios, obesity, and type 2 diabetes to study the complex interactions that influence fracture healing. In my opinion, the paper makes a significant contribution to the field of diabetic bone healing models in mice by critically evaluating existing models, categorizing them, and recommending a high-fat diet model. It provides insight into the complexity of diabetic fracture healing and emphasizes the importance of selecting appropriate models that mimic clinical scenarios. The categorization and comparison of different models provides valuable guidance to researchers in selecting the most appropriate model for their studies. From a technical point of view, this paper is well organized, providing a clear and concise introduction, a detailed categorization of models, a thorough review of the existing literature and a conclusion. The authors describe each model in detail, discussing variations in doses, strains, and diets. The inclusion of tables and figures helps to present the data effectively. In my opinion, this article appears to be scientifically sound and presents relevant information on diabetic bone healing models. The authors discuss potential limitations, such as the influence of leptin knockout in transgenic models and demonstrate a critical understanding of the intricacies of the models. The inclusion of various studies strengthens the scientific basis of the paper. The paper appropriately references previous work, especially when discussing different models. The authors acknowledge the limitations of certain models and provide a nuanced perspective on the challenges of studying diabetic fracture healing. The English used in the text is generally correct and readable and is suitable for a scientific audience.

Suggestions for improvement:

Clarification on model selection: While the paper suggests a high-fat diet model as the most appropriate, more explicit criteria for model selection based on specific research objectives would strengthen the recommendations. Clarity of recommendations: The paper could benefit from a clearer presentation of recommendations for researchers, outlining key considerations when selecting a model.

In conclusion, the paper is a valuable contribution to the field, but minor improvements could further enhance its quality.

Minor corrections:

1.      Figure 2 is not cited in the text.

2.      There is a misspelling “destruction” in the figure 4.

Reviewer 2 Report

Comments and Suggestions for Authors

TITLE:  Diabetic bone healing models in the mouse – advantages and limitations

biomedicines-2717340-peer-review-v1

The aim of the present investigation was to investigate mouse models used to study diabetic fracture healing and evaluates the consistency with human diabetic conditions.

GENERAL COMMENTS

The article is in-line with the journal topic, but flaws should be improved.  The investigation is interesting, and the present paper is recommended for publication to the present journal after major revision.

Title: The title should indicate the type of study that has been conducted: (f.e.: Diabetic bone healing models in the mouse – advantages and limitations: a systematic review)

Introduction

1.     Line 31-43: The introduction is too generic, and the epidemiological data could be removed in favor of a more detailed overview of the diabetes physiopathology and the implications in wound healing.

2.     The different models of bone fracture and defects in animal studies should be well-described in this section, not only describing the long bone model.

Materials and methods

1.     The authors purposed a mix of narrative review with a non-complete systematic review process. In my opinion the authors could maintain the systematic review methodology with some improvements: PROSPERO registration, conformity with PRISMA guidelines, risk of bias assessment.

2.     The search strategy should be revised. The authors purposed an unusual double keyword search for articles screening.

3.     The data summary methods should be described in detail.

4.     How many operators performed the articles screening? Inclusion/exclusion criteria?

Results

The results presentation is a little bit poor. In my opinion the paper quality could be significantly improved describing the general characteristics of the studies included.

The authors could introduce a table summarizing the study data of the papers included in the present review.

Discussion

A separated section for discussions is strongly recommended. The section is lacking regarding the healing time for bone fracture models comparing it with diabetic vs. non-diabetic mouse model. In my opinion, this aspect plays a key role for the review findings.

In addition, the authors could discuss the influence of the diabetic mouse model also for effects size and population sampling for pre-clinical study.

The limits of the present review analysis should be described.

Round 2

Reviewer 2 Report

Comments and Suggestions for Authors

The authors replied to the comments improving significantly the paper presentation quality. The paper is recommended for publication in present form